# Study on Spacing Regulation and Separation Performance of Nanofiltration Membranes of GO

**DOI:** 10.3390/membranes12080803

**Published:** 2022-08-19

**Authors:** Na Meng, Pinping Zhao, Wei Zhou, Jie Yan, Die Hu, Yanqing Fang, Jun Lu, Qiang Liu

**Affiliations:** Jiangsu Key Laboratory of Industrial Pollution Control and Resource Reuse, School of Environment Engineering, Xuzhou University of Technology, Xuzhou 221116, China

**Keywords:** graphene oxide, layer-spacing control, nanofiltration membrane, separation of function

## Abstract

Graphene oxide (GO) membranes have attracted significant attention in the field of water processing in recent years due to their unique characteristics. However, few reports focus on both membrane stability and the “trade-off” effect. In this study, a series of aliphatic diamines (1, 2-ethylenediamine, 1, 4-butanediamine, and 1, 6-hexamethylenediamine) of covalent crosslinked GO were used to prepare diamine-modified nanofiltration membranes, BPPO/AX-GO, with adjustable layer spacing using the vacuum extraction–filtration method. Moreover, Ax-GO-modified nanofiltration membranes modified with adipose diamine had higher layer spacing, lower mass-transfer resistance, and better stability. When the number of carbon atoms was 5, the best layer spacing was reached, and when the number of carbon atoms was greater than 4, the modified membrane nanosheets more easily accumulated. With the increase in layer spacing, the water flux of the composite film increased to 26.27 L/m^2^·h·bar. Meanwhile, adipose diamine crosslinking significantly improved the stability of GO films. The interception sequence of different valence salts in the composite membrane was NaCl > Na_2_SO_4_ > MgSO_4_, and the rejection rate of bivalent salts was higher than that of monovalent salts. The results can provide some experimental basis and research ideas for overcoming the “trade-off” effect of a lamellar GO membrane.

## 1. Introduction

In the 21st century, the shortage of fresh water resources and water pollution have become the key factors restricting development [1]. Membrane-separation technology plays an important role in seawater desalination and in the treatment of polluted water by virtue of its unique advantages, and it shows good application prospects [2].

Generally speaking, when a water source contains various types of pollutants and complex properties, but in relatively low concentrations, it is usually called slightly polluted water [3]. The effects of conventional water-treatment processes are very poor, and chlorine disinfection produces “three substances;” therefore, it is difficult to meet the national drinking-water quality standards [4]. At present, the water quality purification of slightly polluted water is mainly achieved with the following treatments: biological pretreatment [5], chemical-oxidation pretreatment [6], conventional strengthening treatment [7] (enhanced coagulation, enhanced filtration), advanced granular activated carbon treatment [8], biological activated carbon treatment [9], ozone activated carbon treatment [10], photocatalytic oxidation [11], etc.; however, there are certain problems in the practical application of each process [12]. Membrane separation technology has become a research hotspot [13], as it is an effective way to treat micro-polluted water because of its positive water purification effect and its complete removal of organic matter [14]. Nanofiltration membrane technology has become the preferred membrane separation technology for the treatment of micro-polluted water due to its strong selective-separation ability [15]; the removal of disinfection by-products (trace weeding and pesticides, heavy metals, partial hardness), low operating pressure, partial removal of solute ions, excellent resulting water quality, and low energy consumption [16].

In recent years, membranes based on graphene and its derivative, graphene oxide (GO), have been increasingly used as new two-dimensional nanomaterials due to their surface functional groups being easy to modify and their unique flake-like structure [17,18,19]. The GO membrane separation properties and the interlayer are closely connected to the size of the nanometer channel [20]. Due to the GO portion of oxygen-containing functional groups located on the layer with a large number of reactive sites [21], the GO at the edge of the lamellar carboxyl can interact with the amino group of the acylation reaction and with amine substances to form a C-N covalent bond [22]; this feature provides an amine substance with the ability to crosslink GO [23]. Lu [24] et al.’s experiment proved the possibility of using polyethyleneimines of different molecular weights (PEI-600 and PEI-10000) as crosslinking agents to produce GO films with significantly increased layer spacing, compared with the previous ones obtained using the vacuum extraction–filtration method. Yang [23] et al. chose thiourea (TU) as the crosslinking agent, and covalently cross-linked GO using a dehydration–condensation reaction between -NH2 in thiourea molecules or between -SH and -COOH on GO lamellae and the nucleophilic addition reaction between -NH2 and C-O-C. The prepared TU-GO films had better interlayer spacing and regular interlayer channels, and the rejection rate of different salt molecules can reach over 99%, which is determined to be the excellent alcohol–water separation performance [23]. In addition, Liu [25] et al. took advantage of the crosslinking properties of metal cations Fe^3+^ and Al^3+^ to GO films, finding that Fe^3+^ and Al^3+^ could effectively connect adjacent GO lamellae through electrostatic attraction, as well as coordination with GO. Compared with Al^3+^ cross-linked GO films, Fe^3+^ cross-linked GO films had larger interlayer spacing and higher flux [25]. Liu [26] et al. prepared GOCN composite films with high throughput by embedding g-C_3_N_4_ nanosheets into GO films. The g-C_3_N_4_ nanosheets could be evenly distributed among GO layers, and the embedding of g-C_3_N_4_ nanosheets produced more nanochannels in the film in comparison with pure GO films.

Despite the fact that the “trade-off” effect has been extensively reported in studies on the preparation and application of GO separation membranes [27], few reports focus on both membrane stability and the “trade-off” effect. As the flux increases, the rejection rate of various ions and molecules decreases, making it difficult to achieve the coordinated improvement of flux and rejection rate [20]. Gao [28] et al. selected sodium citrate as the reducing agent and prepared GO/Ag NP composite materials by reducing silver nitrate at high temperatures; then, they prepared GO/Ag NP nanofiltration membranes using vacuum extraction. The experiment showed that the membrane stability was improved, but the permeability greatly decreased due to the increase in membrane thickness. In order to improve membrane permeability without reducing the rejection rate, Fang [29] et al. prepared a GO/MWCNT-modified NF membrane by combining MWCNTs and GO. Similarly, Wang [27] et al. used the vacuum assisted self-assembly method to assemble EDA-pre-crosslinked GO nanosheets on tubular ceramic substrates. The results showed that the prepared membrane had a good penetration flux and showed no significant changes in the interception effect, which alleviated the influence of the “trade-off” effect, to a certain extent. In addition, due to the deprotonation of carboxyl groups on GO sheets in an aqueous solution, GO sheets are usually negatively charged, and adjacent GO sheets repel each other [30]. As a result, the membrane is unstable in the water environment and is easily damaged by water washing [31]. Nevertheless, few reports focus on both membrane stability and the “trade-off” effect. Thus, it is urgent to explore the “trade-off” effect of composite GO membranes on the premise of maintaining the stability of the composite membrane.

In the present study, a simple method was developed to adjust the spacing between nanofiltration membranes. In this work, a BPPO membrane was used as the basement membrane, and then a GO separation layer was constructed on it, and adipose diamine was used to adjust the layer spacing of the membrane. This study investigated the effect of layer spacing regulation on the separation performance of nanofiltration membranes, while maintaining the stability of composite membranes, which can provide some experimental basis and research ideas for overcoming the “trade-off” effect of a lamellar GO membrane.

## 2. Experimental Section

### 2.1. Experimental Materials and Reagents

Graphene oxide (GO) was provided by Jiangsu Xianfeng Nanomaterials Technology Co., Ltd., Nanjing, China; brominated poly (phenylene oxide) (BPPO), 1-methyl-2-pyrrolidone(NMP), 1, 2-ethylenediamine, 1, 4-butanediamine and 1, and 6-hexamethylenediamine were provided by Shanghai Aladdin Biochemical Technology Co., Ltd., Shanghai, China; sodium chloride (NaCl), sodium sulfate (Na_2_SO_4_), and magnesium sulfate (MgSO_4_) were provided by Tianjin Fuchen Chemical Reagent Factory, Tianjin, China.

### 2.2. Modified-BPPO/GO-Membrane Synthesis Method

#### 2.2.1. Preparation of BPPO Membranes

BPPO membranes were prepared using phase inversion in a water coagulation bath. Using NMP as an organic solvent, we dissolved 4 g of BPPO in 16 g of NMP to obtain 20 wt% casting solution. Then, the mixture was stirred with a magnetic mixer at room temperature for 24 h. The BPPO/NMP solution was left standing to remove bubbles, and the transparent liquid film with uniform thickness was scraped out with an adjustable micrometer film scraper (thickness: 300 µm); then, the film was immersed in a water coagulation bath. After standing for 24 h, the membrane was transferred to fresh deionized (DI) water and stored for future use.

#### 2.2.2. Synthesis of Adipose-Diamine-Modified GO

Adipose diamine (EDA, BDA, or HDA) was dissolved in a suspension to prepare the Ax-GO solution; then, three modified suspensions were produced: GO-EDA, GO-BDA, and GO-HDA.

In each case, 40 mL of GO aqueous dispersion and the corresponding adipose diamine were mixed, and 50 mL deionized (DI) water was added; then, an ultrasonic bath was conducted at 25 °C for 20 min to obtain a uniform ax-GO solution, which was used for the preparation and characterization of the membranes.

#### 2.2.3. Preparation of BPPO/AX-GO Composite Membranes

The AX-GO solution obtained in the above experiments was filtered onto the BPPO membrane by vacuum filtration. After that, the mixed membranes were placed in a water bath and heated with steam at 70 °C for 3 h. Finally, we put the mixed membranes into petri dishes and added deionized water to keep them wet and preserved. The GO BPPO/AX-GO membranes were named according to the composition of the crosslinking agent; for example, BPPO/EDA-GO means that the cross-linker of the membrane is EDA.

### 2.3. Characterization of BPPO/AX-GO Composite Membranes

The surface composition of the composite membranes was characterized using a Fourier-transform infrared spectrometer (FTIR; Thermo Fisher Scientific, New York, NY, USA). The morphology and microstructure of the composite membranes were analyzed using scanning electron microscopy (SEM; Thermo Fisher Scientific, New York, NY, USA), X-ray diffraction (XRD; Thermo Fisher Scientific, New York, NY, USA) and atomic-force microscopy (AFM; Bruker Nano Lnc, Washington DC, USA).

### 2.4. Membrane-Performance Evaluation

#### 2.4.1. Membrane Flux

Using DI as a raw material, the permeability of the BPPO/AX-GO membranes was evaluated by measuring the membrane flux. For flux tests, five samples of various membranes were tested under the same conditions, and their average values were calculated. The experiment was conducted with a dead-end filtration device consisting of an automatic nitrogen generator, a digital balance, a sealable tank, and a filtration chamber.

The detailed filtration process was as follows: (1) The membrane was prepressed at a transmembrane pressure of 2 bar for 2–3 h to obtain a stable flux. (2) When the flux was stable, the transmembrane pressure was reduced to 1 bar, and the pure water flux was recorded every 30 s; at least 50 measured values were collected, and the average flux value was taken. The membrane flux calculation formula is as follows:(1)$$J=\frac{V}{T\times A}{(L/m}^{2}\cdot h)$$
where J is the membrane flux (L/m^2^·h), V is the filtration volume (L), T is the filtration time (h), and A is the membrane effective area (m^2^).

#### 2.4.2. Membrane Rejection

NaCl, Na_2_SO_4_, and MgSO_4_ solutions at the concentration of 1000 ppm were used as feed solutions to test the retention performance of the membrane. For the rejection test, three samples of various membranes were tested under the same conditions, and their average values were calculated. In the experiment, a dead-end filtration device was used to determine the rejection rate. The calculation formula is as follows:(2)$$R=\frac{C_{0}-C_{1}}{C_{0}}\times 100\%$$
where R is the membrane rejection, C_0_ is the raw liquid concentration, and C_1_ is the filtrate concentration.

The concentration of the diluted solution of the inorganic salt was measured using the conductivity method. For diluted solutions with a single strong electrolyte, the conductivity is proportional to the concentration:(3)$$\bar{L}=\lambda \cdot C$$
where L is the conductivity, C is the liquid concentration, and λ is the equivalent conductivity.

## 3. Results and Discussion

### 3.1. Microstructure of Composite Membranes

#### 3.1.1. Scanning Electron Microscopy (SEM)

In this study, scanning electron microscopy (SEM) was used to characterize the basic morphology of the membranes. The effects of different adipose diamines on the surface morphology of the GO nanofiltration membranes are shown in Figure 1. As can be seen from the figure, the surfaces of the prepared BPPO membrane and BPPO/GO-modified nanofiltration membranes presented a large number of uniform pores, which were relatively smooth and had a dense cortex layer. Compared with the BPPO membrane, the surfaces of BPPO/EDA-GO-, BPPO/BDA-GO-, and BPPO/HDA-GO-modified nanofiltration membranes showed typical wavy wrinkles, no pore structure, and no obvious defects in the membrane. The scanning electron microscopy (SEM) images of the surface of the Ax-GO-modified nanofiltration membranes showed that all Ax-GO-modified nanofiltration membranes were completely covered by a layer of GO nanosheets, and there were no gaps between the GO and BPPO layers, indicating that GO and BPPO were closely bonded via fatty-diamine crosslinking.

#### 3.1.2. Atomic Force Microscopy (AFM)

In order to further study the effect of different adipose diamines on the surface microscopic morphology of GO nanofiltration membranes, the roughness was characterized with AFM, and the results are shown in Figure 2. The study of the roughness of the membrane surface revealed the following characteristics: large roughness, large membrane surface area, many water channels, and large membrane flux. However, it was previously found that when membranes are positioned in a low-lying location, they are more likely to accumulate pollutants that block the membrane pores, reduce the water transport channels, and reduce the membrane flux [32]. It can be seen that, according to the arithmetic average roughness and root mean square roughness (Table 1), the surface of the BPPO membrane was the smoothest, and the surface roughness of BPPO/BDA-GO was the largest, with little difference among BPPO/GO, BPPO/EDA-GO, and BPPO/HDA-GO. In the subsequent experiments, it was seen that the BPPO membrane had the maximum accessibility, while other modified composite membranes had small flux. The reason may be that there were more pollutants on the surface of the modified composite membrane, blocking the aperture, while the BPPO membrane may not be easily contaminated by pollutants.

### 3.2. GO-Nanofiltration-Membrane Spacing

In this experiment, the composite membranes in the dry state were used for XRD tests, and the XRD images of the BPPO/GO composite membranes and the Ax-GO composite nanofiltration membranes were obtained, as shown in Figure 3.

To calculate the distance between film layers, Bragg’s diffraction formula can be used, as shown in Equation (3):(4)$$d=\frac{\lambda }{2\sin\theta }$$
where d is the GO layer spacing, λ is the X-ray wavelength, and θ is the position of the diffraction peak.

Figure 3 shows that the 2θ values were 10.44°, 10.33°, 8.36°, and 7.52°, respectively. According to Bragg’s formula calculations, it could be concluded that the layer-spacing values of the BPPO/GO composite membrane and the BPPO/EDA-GO, BPPO/BDA-GO, and BPPO/HDA-GO composite nanofiltration membranes were 8.5 Å, 8.6 Å, 10.6 Å, and 11.8 Å, respectively. The measured XRD 2θ values were converted to layer spacing, as shown in Figure 4.

With the increase in 2θ, the layer distance gradually increased, and the percentages of increase under the three crosslinking agents were 1.1%, 24.7%, and 38.8%, respectively. According to the previous literature, the change in the interlayer distance under wet conditions is better than that under dry conditions. The hydrogen bond interacts with the π–π bond in the GO layer when the hydrogen bond is broken, and the distance between the layers is stretched. The effective pore size of GO is the channel between the diameter of the water molecules and the hydration ions in the brine. Therefore, it can be determined that the effective use of a crosslinker can meet the adjustment of the distance between different composite membranes. Molecules are inserted into the GO layer to adjust the effective aperture of the GO membrane to determine the best value of GO layer spacing. The effective aperture should be less than or equal to the minimum size of the hydration ions, such as Na^+^ or Cl^−^, that is, less than or equal to 6.6 Å. With optimal layer spacing, the modified composite membrane can show a high flux and high rejection rate [33].

After crosslinking with a diamine monomer, the interlayer carbon chain of the modified GO membrane was determined to be of three types: parallel, vertical, and inclined. If the type of carbon chain is parallel to the GO layer, the layer spacing of the modified GO membrane is independent of the chain length. Based on the previous literature, the equation can be modified as:(5)L=1.5+1.565n
where n is the number of carbons in aliphatic diamines, and L is the chain length of aliphatic diamines.

Correspondingly, when the effective aperture of GO expands to 6.6 Å, the equation is as follows:(6)$$L\cdot \sin\;(54^{{}^{\circ}})\;=(1.5+1.265n)\cdot \sin\;(54^{{}^{\circ}})=6.6$$

Therefore, n ≈ 5.3 = 5. This shows that the optimal carbon number of the chain was 5. So, the GO membrane modified with 1, 5-diaminopentane had the highest water flux and a good ion-retention rate. However, long-chain alkyl diamines easily produce hydrogen bonds when used as crosslinking agents, and the Ax-GO solution has generally poor dispersion. Therefore, when the carbon atom number of the aliphatic terminal diamine was greater than 4, Ax-GO nanosheets were more likely to aggregate.

### 3.3. Fourier-Transform Infrared Characterization

Fourier-transform infrared spectroscopy (FTIR) was used to characterize the chemical reactions between GO and EDA, and BDA and HDA, and the results are shown in Figure 5. The characteristic peak of GO is a carboxyl group. After GO was modified with different adipose diamines, the absorption peaks of C=O and C-O-C on the GO surface disappeared, indicating that the carboxyl and epoxy groups of GO disappeared in the reduction reaction [34]. At the same time, a new band was found at 2941 cm^−1^ of the tensile characteristic peak of the N-H amide bond, and the shoulder peak at 2941~2850 cm^−1^ was caused by antisymmetric tensile vibration. At the same time, new peaks were generated at 1472 cm^−1^ and 820 cm^−1^, corresponding to C-N stretching and CONH deformation vibrations. On the basis of bond disappearance and formation, an amidation reaction may have occurred between the adipose diamine and the GO nanosheets via a nucleophilic addition reaction [35].

### 3.4. Membrane Flux and Rejection

#### 3.4.1. Membrane Flux

In this experiment, the membrane flux and salt-retention ability of the modified BPPO/AX-GO composite membranes were tested using the dead-end device under 1 bar of pressure. The measured membrane flux data are shown in Figure 6. As can be seen from the figure, the maximum water flux of the BPPO membrane was 267.46 L/m^2^·h·bar, while the flux of the BPPO/GO membrane was 17.22 L/m^2^·h·bar. Compared with BPPO, the pure water flux of the mixed membrane with GO was significantly increased, because under a higher GO load, a thicker GO layer has higher resistance to water transportation [36]. Compared with the BPPO/GO membrane without adipose diamine, the water flux of the BPPO/AX-GO modified composite membranes with adipose diamine was increased; the BPPO/EDA-GO and BPPO/BDA-GO membranes showed values of 25.17 L/m^2^·h·bar and 26.27 L/m^2^·h·bar, respectively. The increase in water flux was due to the introduction of adipose diamine monomer, which increased the lamellar spacing of GO film. The increased interlayer nanochannels on the one hand accelerated the penetration of water molecules, but also weakened the sieving effect of the nanochannels on the molecules, leading to the increase in water flux in the composite nanofiltration membrane. Secondly, it can be seen from the above that with the addition of the adipose diamine monomer, the membrane surface roughness increased, the rough nanofiltration membrane surface can provide more water contact sites, and the membrane surface can absorb more water molecules, thus helping to improve the flux of the nanofiltration membrane. Among the membranes, the BPPO/BDA-GO-modified nanofiltration membranes showed the smallest change. It could be seen that the separation effect of BPPO/BDA-GO-modified nanofiltration membranes was relatively stable.

#### 3.4.2. Rejection

Figure 7 shows the separation-performance test results of the BPPO/AX-GO-modified nanofiltration membranes for different salt solutions. The results showed that the BPPO basal membrane had almost no salt-repulsive behavior. Due to the low GO content in this experiment, the rejection rate was generally small [37]. As can be seen from the figure, the rejection rate of the BPPO membrane for the three salt solutions was very low, and the rejection rate for MgSO_4_ was negligible. However, the BPPO/EDA-GO membrane had a relatively good interception ability, because the addition of adipose diamines changed the structure of the GO lamella, changed the distance between layers, improved the size of the interlayer channel, and improved the interception rate. The rejection rates of NaCl, Na_2_SO_4_, and MgSO_4_ were 11.97%, 6.33%, and 2.6%, respectively, showing that NaCl > Na_2_SO_4_ > MgSO_4_. According to the previous literature, the content of GO plays a certain role in the interception rate of salt solutions [38]. When the content of GO is very low, the flux is large, but the interception rate is small [36]. It was found that the membrane flux had no direct influence on the rejection rate. The data showed that with the three salt solutions, the total rejection rate could be expressed as NaCl > Na_2_SO_4_ > MgSO_4_. Previous studies found that salts with divalent anions had higher rejection rates than those with monovalent anions. If the membrane with a negative surface charge has the same cation salt (Na^+^), the rejection rate of divalent anionic salt is higher. In this experiment, the overall rejection rate of the BPPO/EDA-GO-modified nanofiltration membrane was higher than that of the BPPO/BDA-GO-modified nanofiltration membrane, and the rejection rate of the BPPO membrane was the lowest. It may be that the use of crosslinking agents changed the structure of the GO lamella, changed the distance between layers, and improved the size of the layer channel. BDA was better than EDA in regulating the size, and the rejection rate was higher.

## 4. Conclusions

In this paper, the regulation of the nanofiltration-membrane spacing and the separation performance of GO were preliminarily studied. The vacuum extraction–filtration method was adopted to prepare BPPO/GO-modified composite membranes by adding three binary amine molecules as crosslinking agents. The crosslinking effects and performance of the three crosslinking agents were studied, and the following conclusions were drawn:

The use of crosslinkers effectively connected the GO lamellar structure, and different crosslinkers could be used to construct membranes with different layer spacing. In this paper, adipose diamine was used as the crosslinking agent. When the number of carbon atoms was 5, the optimal layer spacing was reached, and the modified GO film had the highest water flux, with good ion rejection. When the number of carbon atoms was greater than 4, the modified film more easily aggregated. The Ax-GO nanofiltration membranes modified with adipose diamine showed better stability, with a reduced 2θ value, increased layer spacing, and reduced mass-transfer resistance. Moreover, the permeation flux of the membrane increases.

The size of the membrane flux was related to the size of the membrane aperture, or the distance between membrane layers. Via membrane modification, the flux of the BPPO/GO membrane was significantly improved. From 17.22 L/m^2^·h·bar for the BPPO/GO membrane, the value increased to 25.12 L/m^2^·h·bar for the BPPO/EDA-GO membrane and to 26.27 L/m^2^·h·bar for the BPPO/BDA-GO membrane, which increased by about 50%. The increase in water flux was due to the introduction of the adipose diamine monomer, which increased the lamellar spacing of the GO film. The increased interlayer nanochannels accelerated the penetration of water molecules, but also weakened the sieving effect of the nanochannels on the molecules, leading to the increase in water flux in the composite nanofiltration membrane. On other hand, it can be seen from the above that with the addition of the adipose diamine monomer, the membrane surface roughness increased, the rough nanofiltration membrane surface provided more water contact sites, and the membrane surface could absorb more water molecules, thus helping to improve the flux of the nanofiltration membrane.

It was found that the interception order of the Ax-GO nanofiltration membranes modified with adipose diamines for different valence salts was NaCl > Na_2_SO_4_ > MgSO_4_. The rejection rate of divalent salts was higher than that of monovalent salts, which was caused by the sieving effect and the Donnan effect.

## Figures and Tables

**Figure 1 membranes-12-00803-f001:**
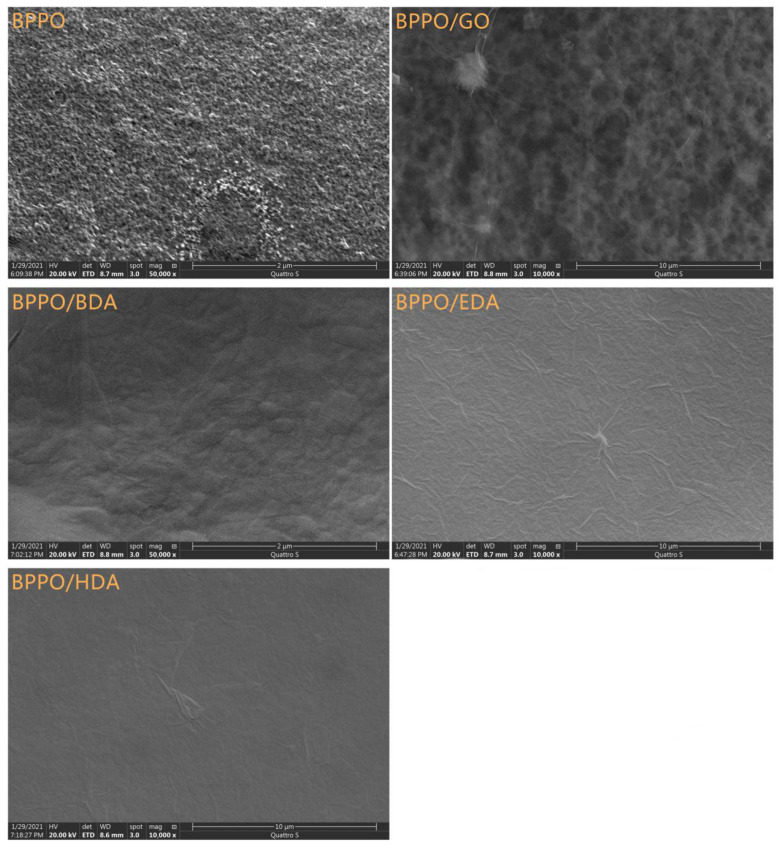
SEM images of the BPPO membrane, the BPPO/GO membrane, and the Ax-GO-modified nanofiltration membranes.

**Figure 2 membranes-12-00803-f002:**
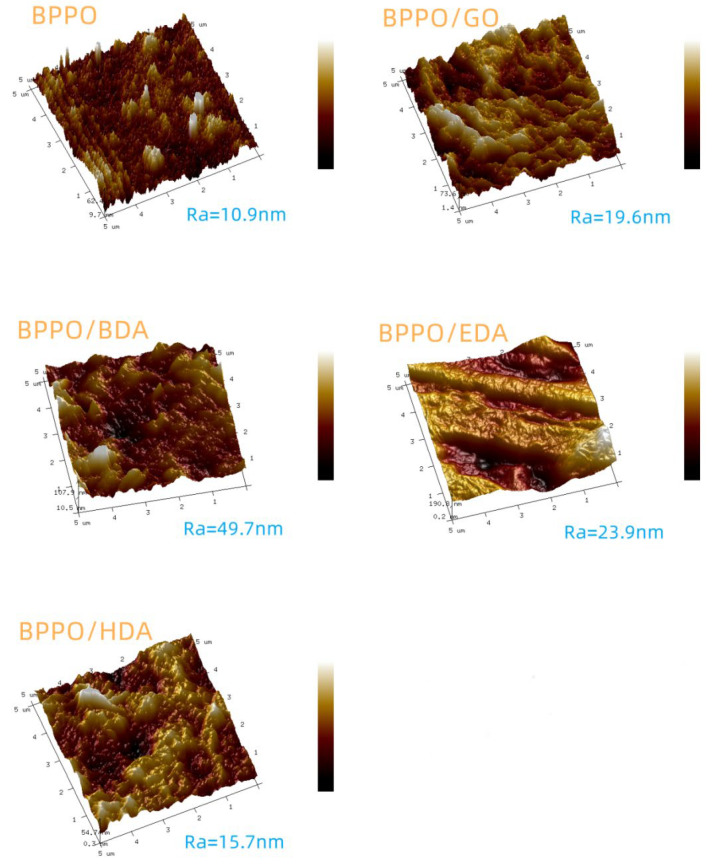
AFM images of the BPPO membrane, BPPO/GO membrane, and Ax-GO-modified nanofiltration membranes.

**Figure 3 membranes-12-00803-f003:**
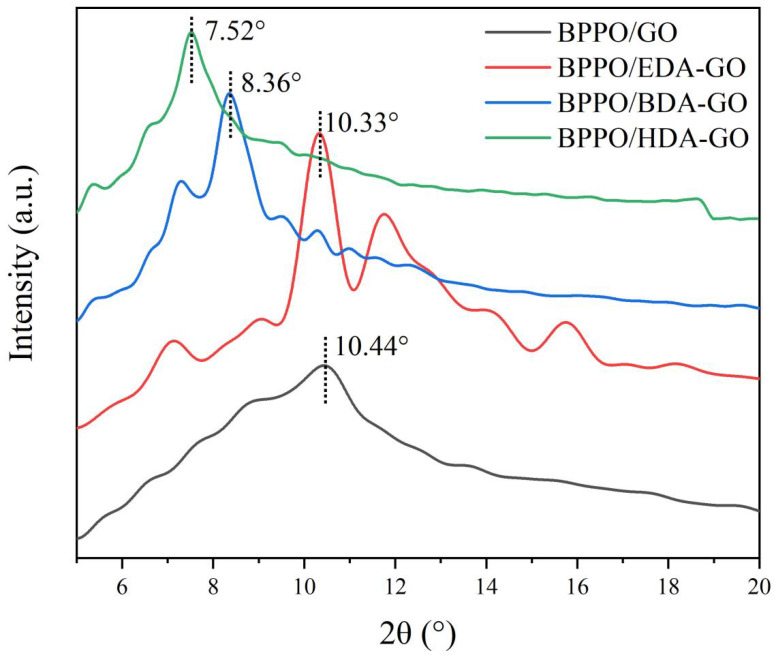
XRD patterns of BPPO/GO and Ax-GO-modified membranes.

**Figure 4 membranes-12-00803-f004:**
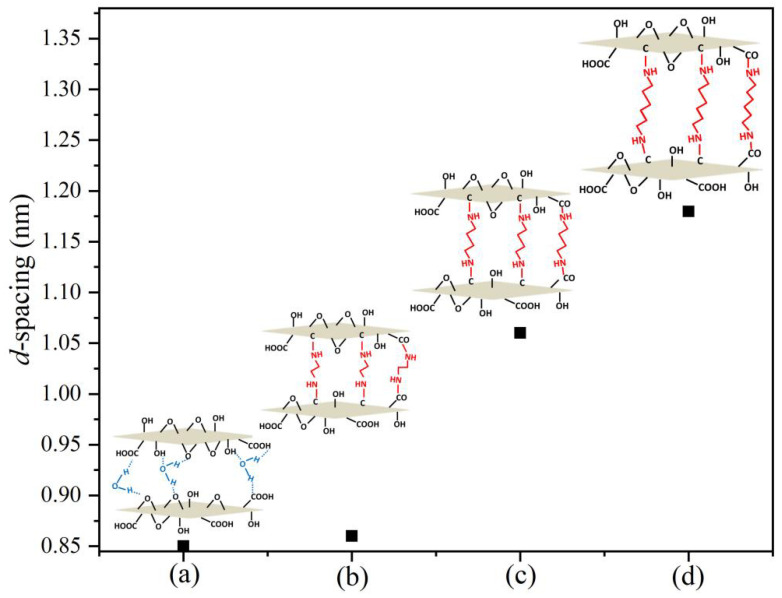
Change of spacing between BPPO/GO and three composite membranes: (**a**) BPPO/GO; (**b**) BPPO/EDA-GO; (**c**) BPPO/BDA-GO; (**d**) BPPO/HDA-GO.

**Figure 5 membranes-12-00803-f005:**
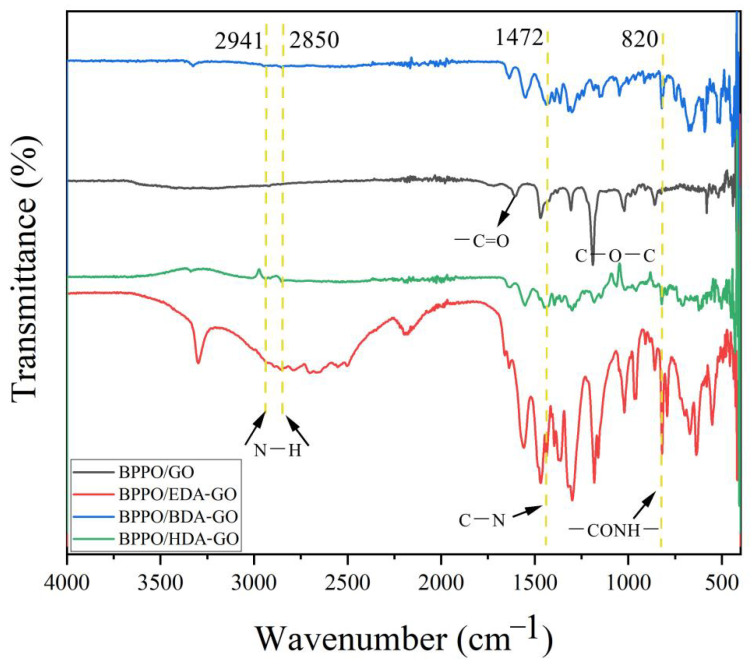
FTIR spectra of the BPPO/GO membrane and the Ax-GO-modified nanofiltration membranes.

**Figure 6 membranes-12-00803-f006:**
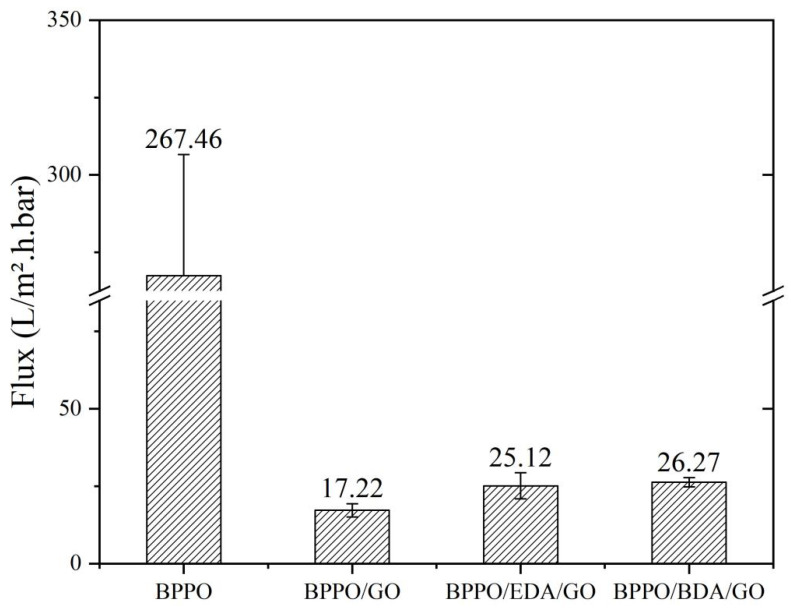
Flux of BPPO/AX-GO-modified nanofiltration membranes.

**Figure 7 membranes-12-00803-f007:**
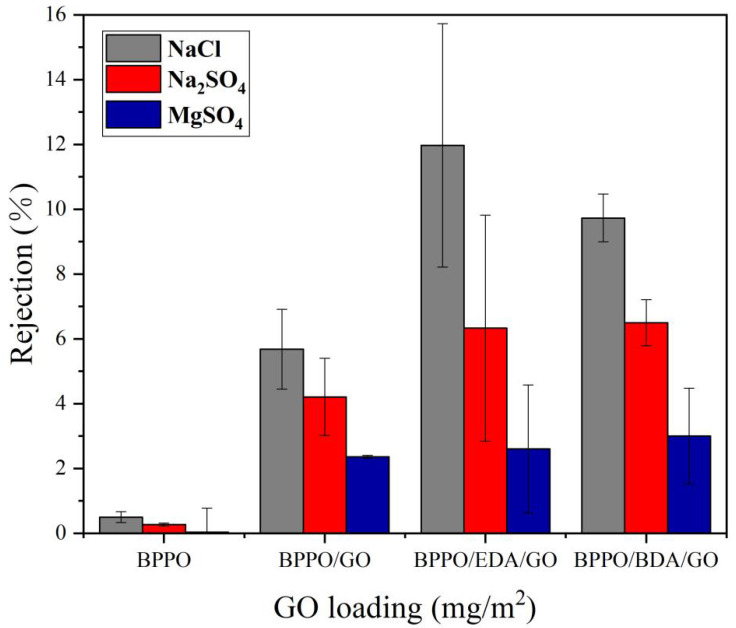
Influence of BPPO/ AX-GO-modified nanofiltration membranes on rejection rates.

**Table 1 membranes-12-00803-t001:** Roughness average and root mean square of the BPPO membrane, BPPO/GO membrane, and Ax-GO-modified nanofiltration membranes.

Sample	Mean Roughness/nm	Root Mean Square Roughness/nm
BPPO	10.9	14.2
BPPO/GO	19.6	24.5
BPPO/BDA-GO	49.7	56.8
BPPO/EDA-GO	23.9	29.6
BPPO/HDA-GO	15.7	22.5

## Data Availability

The data presented in this study are available on request from the corresponding author.

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
