# Peer review of "Study on Spacing Regulation and Separation Performance of Nanofiltration Membranes of GO"

_membranes, 2022, doi:10.3390/membranes12080803_

Round 1
Reviewer 1 Report
The whole paper generally has been written and organized in a good way. However, some points need to be addressed:
1- The novelty, contribution and research gap need to be highlighted and well presented in introduction. You may want to use the following ref:
https://doi.org/10.1016/j.desal.2021.115539
2- Discussion should be more scientifically explained. Why GO? This should be explained.
3- Conclusion part should be covering more finding.
4- English needs a polishing.
Reviewer 2 Report
The current manuscript describes the “Study on Spacing Regulation and Separation Performance of Nanofiltration Membrane of Go”. In this study, a series of aliphatic diamines (1, 2‐ethylenediamine, 1, 4‐butanediamine and 1, 6‐hexamethylenediamine) of covalent‐crosslinked GO were used to prepare diamine‐modified nanofiltration membranes. The membrane's performance was studied for the separation of bivalent and monovalent salts. After reviewing the manuscript reviewer pointed out that there is a lot of scope for improvement of this manuscript. A lot of sections may need further illustration; and more importantly, because this type of work has been already reported by a lot of researchers using different types of nanomaterials/nanosheets. One important point is that the abstract need to be revised completely.
The authors should consider critically these comments to improve the quality of the work.
Few examples of where authors should make changes are provided below:
- The abstract is not showing the exact findings and output and what the authors want to show here in short. Authors have to rewrite the abstract.
- In the introduction after reference 23, this sentence the reference needs to be there as per discussion like Chem. Eng. J., 373 (2019) 1190-1202; Comprehensive Analytical Chemistry, 91, 2020, 73-97.
- 4th para needs to rewrite again, Authors have to compare the Go materials effect with the recent literature and cite them for the confirmation of the sentences in this para like Chemical Engineering Journal Volume 446, 2022, 137303; International Journal of Biological Macromolecules Volume 193, 2021, Pages 2121-2139; Journal of Membrane Science Volume 609, 2020, 118212; Journal of Cleaner Production 365 (2022) 132858. etc.
- Section 2.2.3. Preparation of BPPO/ AX‐GO composite membranes needs to write in detail.
- All equations need to check and need the equation number in the manuscript.
- Justify the AFM results with RMS and Ra values in the manuscript.
- Proper labelling and values should be there for figure 6. Particular functionality has to be discussed in the manuscript.
- Section 3.4.1. Membrane flux needs to be discussed more with the proper mechanism.
- If possible justify the pressure effect on the membrane performance.
- Authors have to look always starting words of the sentences.
- Grammar needs to check one more time.
Round 2
Reviewer 1 Report
The comments have been addressed.
Reviewer 2 Report
Authors have significantly addressed the comments so paper can be accepted.